# Outcome of Children and Adolescents with Recurrent Classical Hodgkin Lymphoma: The Italian Experience

**DOI:** 10.3390/cancers14061471

**Published:** 2022-03-13

**Authors:** Alberto Garaventa, Stefano Parodi, Giulia Guerrini, Piero Farruggia, Alessandra Sala, Marta Pillon, Salvatore Buffardi, Francesca Rossi, Maurizio Bianchi, Marco Zecca, Luciana Vinti, Elena Facchini, Tommaso Casini, Sayla Bernasconi, Loredana Amoroso, Salvatore D’Amico, Massimo Provenzi, Raffaela De Santis, Antonella Sau, Paola Muggeo, Rosa Maria Mura, Riccardo Haupt, Maurizio Mascarin, Roberta Burnelli

**Affiliations:** 1Paediatric Oncology Unit, IRCCS Istituto Giannina Gaslini, L.go G. Gaslini 5, 16147 Genova, Italy; albertogaraventa@gaslini.org (A.G.); loredanaamoroso@gaslini.org (L.A.); 2Epidemiology and Biostatistics Unit, IRCCS Istituto Giannina Gaslini, L.go G. Gaslini 5, 16147 Genova, Italy; stefanoparodi@gaslini.org (S.P.); riccardohaupt@gaslini.org (R.H.); 3UOC Pediatria e Neonatologia, Grosseto USL-Toscana Sud-Est, Via Senese, 58100 Grosseto, Italy; giulia.guerrini@uslsudest.toscana.it; 4Paediatric Haematology and Oncology Unit, A.R.N.A.S. Civic Hospital, Piazza Leotta Nicola 4, 90127 Palermo, Italy; piero.farruggia@arnascivico.it; 5Department of Paediatrics, Ospedale San Gerardo, University of Milano-Bicocca, Fondazione MBBM, Via Cadore, 20900 Monza, Italy; ale.sala@asst-monza.it; 6Dipartimento di Oncoematologia Pediatrica, Università di Padova, Via Gattamelata 5687, 35128 Padova, Italy; marta.pillon@unipd.it; 7Paediatric Haemato-Oncology Department, Santobono-Pausilipon Children’s Hospital, Via Mario Fiore 6, 80129 Naples, Italy; salvatorebuffardi@hotmail.it; 8Dipartimento di Pediatria II Ateneo di Napoli, Servizio di Oncologia Pediatrica, Via Luigi De Crecchio 2, 80138 Naples, Italy; francesca.rossi@unicampania.it; 9Pediatric Onco-Hematology and Stem Cell Transplant Division, City of Health and Science, Regina Margherita Children’s Hospital, Piazza Polonia 94, 10126 Turin, Italy; maurizio.bianchi@unito.it; 10Oncoematologia Pediatrica, Fondazione IRCCS Policlinico San Matteo, Viale Camillo Golgi 19, 27100 Pavia, Italy; m.zecca@smatteo.pv.it; 11Department of Hematology/Oncology and Stem Cell Transplantation, Bambino Gesù Children’s Hospital, Piazza di Sant’Onofrio 4, 00165 Rome, Italy; luciana.vinti@opbg.net; 12Pediatric Oncology and Hematology Unit “LallaSeràgnoli”, Department of Pediatrics, University of Bologna, Sant’Orsola Malpighi Hospital, Via Giuseppe Massarenti 9, 40138 Bologna, Italy; elena.facchini@aosp.bo.it; 13Division of Pediatric Oncology/Hematology, Meyer University Children’s Hospital, Via Gaetano Pieraccini 24, 50139 Florence, Italy; tommaso.casini@meyer.it; 14Pediatric Hematology Oncology, Bone Marrow Transplant, S. Chiara University Hospital of Pisa, Via Bonanno Pisano 93, 56126 Pisa, Italy; s.bernasconi@ao-pisa.toscana.it; 15Paediatric Haemato-Oncology Unit, Department of Clinical and Experimental Medicine, University of Catania, Piazza Università 2, 95124 Catania, Italy; sdamico@unict.it; 16Department of Pediatrics, Civic Hospital, Piazza OMS 1, 24127 Bergamo, Italy; mprovenzi@asst-pg23.it; 17IRCCS Casa Sollievo della Sofferenza, San Giovanni Rotondo, Viale Cappuccini, 47156 Foggia, Italy; r.desantis@operapadrepio.it; 18Pediatric Hematology-Oncology Unit, Ospedale Civico, Via Fonte Romana 8, 65124 Pescara, Italy; antonella.sau1@gmail.com; 19Department of Biomedicine of Developmental Age, University of Bari, Piazza Umberto I 1, 70121 Bari, Italy; paola.muggeo@gmail.com; 20Department of Paediatric Oncohaematology, Microcitemico Hospital, Via Edward Jenner 18, 09121 Cagliari, Italy; rosamaria.mura@aob.it; 21AYA Oncology and Pediatric Radiotherapy Unit, CRO-Centro di Riferimento Oncologico di Aviano, IRCCS Aviano, Via Franco Gallini 2, 33081 Aviano, Italy; mascarin@cro.it; 22Pediatric Hemato-Oncology Unit, Azienda Ospedaliero Universitaria Sant’Anna di Ferrara, Cona, Via Aldo Moro, 44124 Ferrara, Italy

**Keywords:** Hodgkin’s lymphoma, children, adolescents, relapse, survival, prognostic factors

## Abstract

**Simple Summary:**

Survival of classical Hodgkin’s lymphoma (cHL) in Western countries is excellent. However, about 10% of patients with stage I–II disease and 15–30% of those with advanced stages require salvage therapy for resistant or relapsing disease. Many studies have investigated prognostic factors in adult patients, but data on children and adolescents are scarce. We analyzed a cohort of 272 patients aged <18 years with recurrent cHL, enrolled in two Italian subsequent protocols between 1996 and 2016. Overall and event-free survival after 10 years since the first recurrence were 65.3% and 53.3%, respectively. Major prognostic risk factors were progressive disease, advanced stage, ≥5 involved sites, and extra-nodal involvement at the recurrence. Patients with progressive disease, advanced stage, or ≥5 involved sites had a very poor survival and might benefit from more innovative approaches since the first progression. Patients who relapsed later with localized cHL might be considered for a conservative approach.

**Abstract:**

The objective of this study was to identify prognostic factors for children and adolescents with relapsed or progressive classical Hodgkin’s lymphoma (cHL) to design salvage therapy tailored to them. We analyzed a homogeneous pediatric population, diagnosed with progressive/relapsed cHL previously enrolled in two subsequent protocols of the Italian Association of Pediatric Hematology and Oncology in the period 1996–2016. There were 272 eligible patients, 17.5% of treated patients with cHL. Overall survival (OS) and event-free survival (EFS) after a 10-year follow-up were 65.3% and 53.3%, respectively. Patients with progressive disease (PD), advanced stage at recurrence, and ≥5 involved sites showed a significantly worse OS. PD, advanced stage, and extra-nodal involvement at recurrence were significantly associated with a poorer EFS. Multivariable analysis identified three categories for OS based on the type of recurrence and number of localizations: PD and ≥5 sites: OS 34%; PD and <5 sites: OS 56.5%; relapses: OS 73.6%. Four categories were obtained for EFS based on the type of recurrence and stage: PD and stage 3–4: EFS 25.5%; PD and stage 1–2: EFS 43%; relapse and stage 3–4: EFS 55.4%; relapse and stage 1–2: EFS 72.1%. Patients with PD, in advanced stage, or with ≥5 involved sites had a very poor survival and they should be considered refractory to first- and second-line standard chemotherapy. Probably, they should be considered for more innovative approaches since the first progression. Conversely, patients who relapsed later with localized disease had a better prognosis, and they could be considered for a conservative approach.

## 1. Introduction

Classical Hodgkin’s lymphoma (cHL) is a common hematological malignancy in children, adolescents, and young adults. In pediatric patients, it is considered one of the most curable neoplastic diseases. The 5-year survival rate of pediatric cHL patients treated with the modern protocols available in Western countries now exceeds 90% [1]. Based on past experience, these protocols are designed to provide at least similar survival rates as in the past, but with lower treatment burden, to minimize the risk of early- and long-term toxicities. Still, 10% of stage I–II patients and 15–30% of those with advanced stages experience resistant or relapsing disease and therefore require salvage therapy [2]. The chance of rescuing these patients is around 50–60%, with different salvage protocols including standard-dose chemotherapy, high-dose chemotherapy with autologous stem cell rescue (ASCR), haploidentical transplant, related or unrelated transplant, and, quite recently, immunotherapy [3,4,5].

Several studies on adult cHL patients investigated the prognostic significance of several risk factors at relapse (i.e., timing, stage, bulky disease, B symptoms) and led to the development of a prognostic index and treatment guidelines [6]. Conversely, few studies focused on children and adolescents with progressive/relapsed cHL, and their unique conclusion was that the interval between diagnosis and recurrence is the only significant factor affecting survival [2,7].

The objective of this study was to identify any other prognostic factor for children with relapsing or progressive cHL, which could contribute to the design of salvage therapy. With this purpose, we analyzed a large homogeneous population of children and adolescents with progressive/relapsed cHL after being enrolled in two subsequent front-line protocols adopted by the Italian Association of Pediatric Hematology and Oncology (AIEOP) in the period 1996–2016 [8].

## 2. Patients and Methods

Eligible patients were diagnosed with cHL, aged <18 years, treated with MH96 protocol (February 1996–May 2004) or LH2004 (June 2004–December 2016) [8], and who experienced progressive disease (PD) or relapse during or after front-line therapy. Inclusion criteria, procedures, staging, and treatment outlines were recently reported [8] and are summarized in Table 1. After completion of treatment, both protocols required minimum 6-month monitoring for the subsequent 5 years.

Tumor regrowth was defined based on its timing during treatment or after the first elective end of treatment (EOT), namely (i) progressive or refractory disease (PD) when tumor regrowth occurs either during treatment or within 3 months after EOT, (ii) early relapse when it occurs between 3 and 12 months after EOT, and (iii) late relapse when it occurs >12 months after EOT [2,7].

For each patient, data were collected on tumor characteristics at diagnosis: gender, age, tumor histology [9], stage, presence of symptoms A or B, bulky disease, number of involved sites, involvement of extra-nodal sites, protocol, and treatment group including radiotherapy. Collected data about tumor regrowth were as follows: timing of regrowth, age, staging at recurrence, number of involved sites, extra-nodal site involvement, recurrence at the same primary site, and recurrence within the radiotherapy field. Since fluorodeoxyglucose positron emission tomography (FDG-PET) scan was not a standard investigation during the study period, the FDG-PET response was not considered as a prognostic factor.

No AIEOP approved salvage protocol was available, and the most common schemes used as second-line therapies [10,11,12] are reported in Table 2. They were followed by radiotherapy and/or high-dose chemotherapy (HDCT) with ASCR. The individual choice was made according to institutional strategies and, mostly, to previous treatment and characteristics of relapse; further modifications might have been influenced by the initial response to salvage therapy.

Follow-up data after recurrence included date and type of any new event—further progression/relapse, subsequent malignant neoplasm (SMN), date, and status at the last follow-up.

Second-line strategies were approved by ethics committee or institutional review board of each participating institution. Written informed consent was obtained from parents or legal guardians of all patients. The data lock point was September 2019 and data on previous relapses were updated in April 2021.

### Statistical Methods

All data were collected in a central database. Overall survival (OS) after the first recurrence was calculated from the date of PD or relapse to the date of either death due to any cause or last contact. Event-free survival (EFS) was calculated from the date of progression/relapse to the date of either first subsequent event (further progression, second relapse, SMN, or death, whichever occurred first) or date of the last follow-up. The Kaplan–Meier method was used to estimate EFS and OS, while differences between groups were assessed using the log-rank test. Univariable and multivariable survival analysis was carried out by the Cox regression model [13].

All analyses were performed by STATA 13.1 (Stata Corporation, College Station, TX, USA). *p*-values < 0.05 were considered as statistically significant.

## 3. Results

Patient selection for data analysis is illustrated in Figure 1. In total, 82 (16.4%) of the 499 cHL patients treated with MH96 protocol and 190 (17.7%) of the 1074 treated with LH2004 protocol experienced either PD or recurrence, for a total of 272 eligible cHL patients.

Patient characteristics at diagnosis are resumed in Table 3. Patients treated with LH2004 protocol were 69.8%. There was a higher number of males and patients <15 years with nodular sclerosis, B symptoms, bulky disease, and treatment group 3. Patient characteristics were quite similar in the two protocols, except for a significant higher number of adolescents 37.9% vs. 20.7%), nodular sclerosis (93.2% vs. 80.5%), and multiple involved sites (≥8 sites: 37.4% vs. 25.6%) in the LH2004 group (Appendix A). Furthermore, 31% of the LH2004 patients did not receive radiotherapy due to PD vs. 12.2% of the MH96 patients. 

Patient characteristics at disease recurrence are reported in Table 4. Forty-three percent had PD. Time from off-therapy to recurrence was reported for non-refractory patients: 51% relapsed between 3 months and 1 year from stop therapy. About half of the patients were ≥15 years. For 34 subjects (12.5%), no information was available about disease stage at relapse. Advanced stages (3 and 4) were observed in about half of patients. The large majority (64.7%) had no extra-nodal involvement and had recurrence in the site of diagnosis (83.1%) and in an irradiated site (83.3%).

A smaller proportion of relapsed patients was observed in the LH2004 group (49.5% vs. 74.4%, Appendix A), which was characterized by a greater number of patients ≥15 years (60.5% vs. 40.2%), multiple involved sites (≥5: 35.7% vs. 10.8%), and recurrence at the same site of diagnosis (87.5% vs. 73.0%).

Overall, 170 patients received HDCT with ASCR as consolidation of a complete response after second-line therapy. Only 18 patients (9 in complete response) received allogeneic transplant.

Since the date of the first cHL recurrence, 158 further events were observed in 121 patients: 64 further cHL relapses, 5 SNMs, and 89 deaths (Appendix A). In details, relapses occurred between 1 and 118 months (median 11 months); SMNs were: one lung adenocarcinoma at 53 months, one soft tissue sarcoma at 85 months after secondary acute myeloblastic leukemia, one osteosarcoma of the chest wall at 64 months, one colon adenocarcinoma associated with a clear-cell renal carcinoma at 182 months, and a papillary and follicular thyroid carcinoma at 125 months in a patient who had experienced a second cHL relapse at 20 months. The 89 deaths happened between 12 days and 128 months (median 19 months): 33 after HL recurrence, 3 after SMN, and 53 because of a persistent HL.

Figure 2 shows the Kaplan–Meier survival curves for the OS and EFS after disease recurrence in the 272 cHL patients under study. After a 10-year follow-up, OS was 65.3% (95% CI: 59–71) and EFS 53.3% (95% CI: 47–59).

Table 5 shows OS after recurrence by patient characteristics at diagnosis. A significantly better result was reported for patients treated with LH2004 protocol. No difference was observed for gender, age, histology, stage, bulky disease, number, and type of involved sites, as well as radiotherapy administration. B symptoms were associated with a poorer survival, but statistical significance was not reached. A not significant better survival was observed in patients without bulky disease and in those in the first treatment group.

Table 6 shows OS after recurrence by patient characteristics at the time of the event. OS was significantly poorer for patients with PD, advanced stage, a large number of involved sites and extra-nodal site involvement (in this latter not statistically significant). No difference was observed between early vs. late relapse.

Appendix A shows EFS after recurrence by patient characteristics at diagnosis. Only the presence of B symptoms was associated with poorer survival (47.7% vs. 61.6%), without reaching the statistical significance.

Table 7 shows EFS after recurrence by patient characteristics at disease recurrence. Progressive disease, advanced stage, and extra-nodal site involvement were significantly associated with a poorer survival. A not statistically significant lower EFS was observed for patients with at least five involved sites and recurrence at previously irradiated sites.

Multivariable analysis (Table 8) confirmed a higher OS for patients treated with LH2004 protocol (HR = 0.43) and a lower OS for those with progressive disease (HR = 0.33) and with a high number of involved sites at recurrence (HR = 2.5 for ≥5 involved sites)

EFS was significantly related to type of recurrence (HR = 0.34 for relapse vs. progression), stage at relapse (HR = 1.7 for advanced vs. not-advanced stages), and recurrence at the same irradiated site (Table 9). The highest survival of patients treated with LH2004 protocol was confirmed (HR = 0.56).

Combining the potential prognostic factors, evaluated by multivariable analysis, three categories were identified for OS (Figure 3A): progressive patients with ≥5 involved sites and the worst survival (34%, 95% CI: 19.6–49), progressive patients with <5 involved sites with intermediate survival (56.5%, 95% CI: 43.2–67.8), and relapsed patients with the best survival (73.6%, 95% CI: 64.7–80.6). The small number of deaths in this latter group prevented to further separate patients by the number of involved sites.

With regard to EFS, type of recurrence combined with stage identified four categories (Figure 3B). The worst EFS was observed for patients with progression in advanced stage (25.5%, 95% CI: 13.8–38.9) and the best for those with relapse in non-advanced stage (72.1%, 95% CI: 59.2–81.5). Analysis restricted to patients with recurrence after radiotherapy (*n* = 174) showed a similar trend (data not shown) and an extremely poor survival in patients in advanced stage, with PD relapsed in an irradiated site (EFS = 9.5%, 95% CI: 1.6–26.1).

## 4. Discussion

Despite improvements in the overall prognosis for children with cHL, a proportion of patients are not cured by their primary treatment. In adults, clinical risk factors for patients with recurrent disease have been well described and they include refractory disease, time to relapse, advanced stage, bulky disease, extra-nodal disease, and age [1,6]. In addition, an interim FDG-PET positivity was found associated with an increased risk of failure [1]. Preliminary data also suggested that cell-free DNA may be an important biomarker of response and outcome [6]. 

Limited data are available in the pediatric population [7,14,15] and pediatric series include few cases collected over a long period of time. The first investigation published in 1992 reported 35 cases failing first-line therapy with 45% OS at 10 years; the authors identified a poor risk group of patients not achieving a complete response to first-line therapy or relapsing within 1 year since diagnosis [15]. Another study analyzed 176 patients in the period 1986–2003 (51 after progression during or shortly after first-line therapy) [7] and, in agreement with our findings, reported that disease refractory to first-line therapy was the strongest prognostic factor. Patients with progression had 41% disease-free survival after 10 years compared with 86% in patients with relapse, although none of them received ASCR in second remission. Friedmann et al. described 64 relapsed patients in the period 1990–2006 focusing on methods of detection and timing of relapse and concluded that frequent imaging does not appear to impact on survival as most relapses are identified through history and physical examination [14]. Unfortunately, this interesting point could not be addressed in our study.

The overall relapse rate of 17.5% observed in our cohort is similar to that reported in other pediatric studies. OS for our patients was 64%, with a median survival time of 88.6 months, and 65.3% of them survived ≥10 years after recurrence. Survival was better for patients recurring after LH2004 protocol, probably due to an improvement of second- and third-line rescue therapy.

The duration of first remission was a highly significant prognostic factor. Children who relapsed before or within 3 months from completion of treatment had significantly poorer EFS and OS when compared to those who relapsed later. However, we did not observe a difference between early and late relapse. It is difficult to explain the difference in PD between the two protocols: in the period of enrolment of LH2004, it seems to occur a more aggressive disease, as suggested by the higher number of involved sites. The more frequent assessment of response and a larger use of FDG-PET at the EOT in LH2004 protocol could also have played a role.

The patients with PD, in stage 3–4 or with ≥5 involved sites had a very poor survival. These patients should be considered refractory to first-line therapy and also to second-line standard chemotherapy. Probably, they should be treated with more innovative approaches since first progression. Conversely, patients who relapsed later on with a localized disease had a better prognosis and they could be considered for a conservative approach. However, in such patients we were unable to identify a more favorable subset only on the basis of timing of relapse. Even if consistent with evidence from other studies, our results of stratified analysis should be considered cautiously due to the adopted post hoc approach, and they need to be validated in a large independent prospective cohort. 

The strength of our study is the huge number of patients treated at diagnosis with only two homogenous protocols, and the consequent large population of relapsed patients. To our knowledge, our study is the largest one in a pediatric population. Limitations include the multicenter retrospective design and the analysis of only static variables, like stage and number of involved sites. Early response to salvage therapy was not available, due to the lack of detailed information on salvage therapies, even though they were likely to be quite homogeneous. However, when designing second-line treatment, early response to salvage therapy should be considered and poor responder patients should be moved to a higher risk group and treated with novel therapies [2,16].

In conclusion, current salvage therapeutic approaches are quite effective, with a 65.3% 10-year OS in cHL patients after progression or relapse. However, identification of several prognostic groups should be advisable to address different therapeutic approaches. Our results, in agreement with previous studies, strongly indicate that refractory disease at the first-line treatment represents the most relevant prognostic factor. Advanced stage at recurrence, the number of involved sites and recurrence at the same previously irradiated district could also significantly affect patient survival. 

Alongside the clinical characteristics, liquid biopsy-based biomarkers, such as cell-free DNA can represent new tools to assess a prognostic stratification at diagnosis. Moreover, it can monitor the quality of response detecting the minimal residual disease, or the recurrence of the disease. These new biomarkers should be considered among the future predictive features, considering the improving technology and the emerging results, even though the requirement of validation in prospective analyses in childhood cHL [17,18,19]. 

Innovative therapy like Brentuximab alone [20] or combined with Nivolumab [2] or with chemotherapy [21] has been studied in refractory or relapsed patients achieving very high overall and complete response rates, and it should be considered pre and/or post HDCT/ASCR in the poor prognosis patients or in patients resistant to second-line chemotherapies. The use of these drugs in the treatment of low- and standard-risk groups could be considered to decrease long-term toxicity in selected patients and in cooperative trials as their long-term side effects are not well known yet.

## 5. Conclusions

The identification of prognostic groups of children and adolescents with refractory or relapsing cHL is needed in order to address different therapeutic approaches. In our study, patients with PD, in advanced stage or with ≥5 involved sites had a very poor survival. They should be considered refractory to first- and second-line standard chemotherapy and could be considered for more innovative approaches since the first progression. On the contrary, patients who relapsed later with localized disease had a very good outcome, and they could be addressed to a more conservative therapeutic approach.

## Figures and Tables

**Figure 1 cancers-14-01471-f001:**
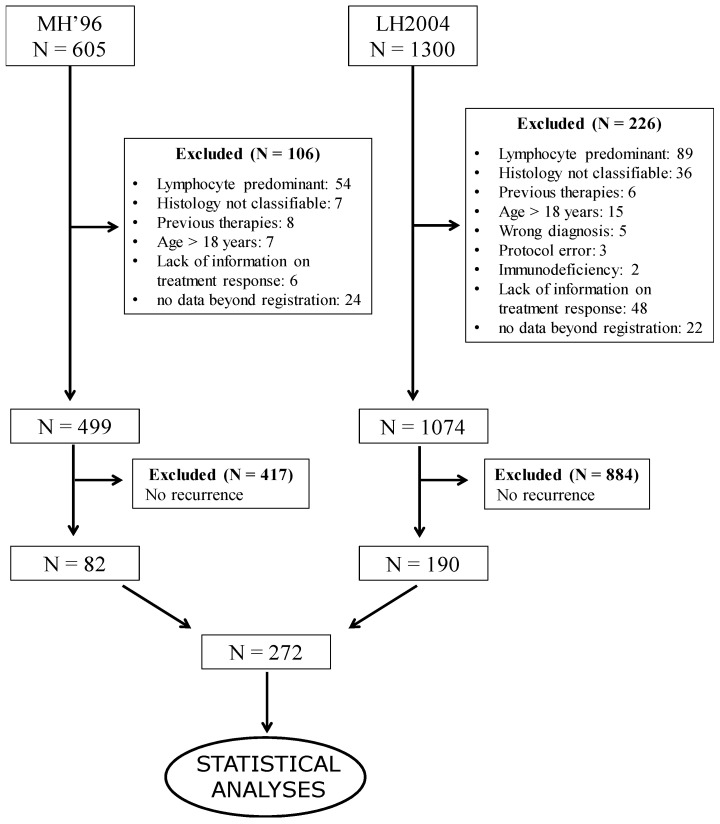
Patient selection for statistical analyses.

**Figure 2 cancers-14-01471-f002:**
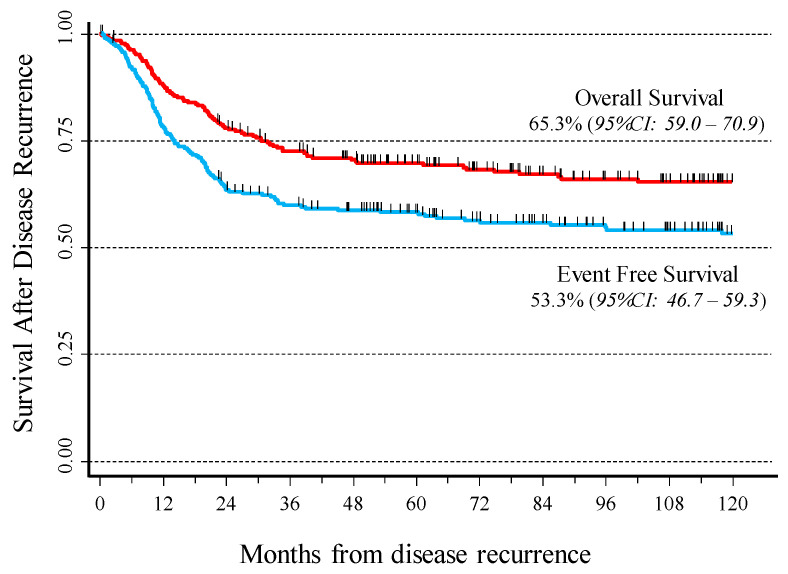
Overall and event-free survival after disease recurrence in 272 cHL patients.

**Figure 3 cancers-14-01471-f003:**
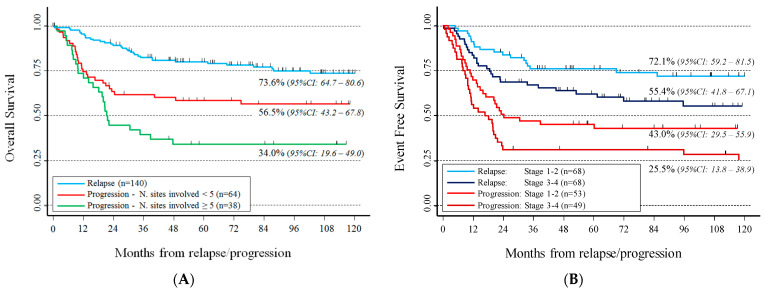
Survival after relapse or progression of 272 classical Hodgkin’s lymphoma patients by characteristics at relapse identified by multivariable Cox regression model. Panel (**A**): overall survival. Panel (**B**): event-free survival.

**Table 1 cancers-14-01471-t001:** Summary of MH96 and LH2004 AIEOP protocols.

Treatment Groups	MH’96	LH2004
1 IA, IIA supradiaphragmatic no bulky, no pulmonary hilum, <4 lymphatic sites, or IA, IIA infradiaphragmatic <4 lymphatic sites	3× ABVD+: - CR no initial mediastinal involvement: stop - others: RT - CR o PR ≥ 75%: 20 Gy IF - PR <75%: 36 Gy IF	3× ABVD+: - CR: stop - others: RT 25.2 Gy IF
2 patients not included in groups 1 or 3	4× COPP/ABV+RT: - CR or PR ≥ 75%: 20 Gy IF - PR < 75%: 36 Gy IF	4× COPP/ABV+: - CR: RT 14.4 Gy IF - PR: 2× IEP + RT - CR: 14.4 Gy IF - PR: 25.2 Gy IF
3 IIIB, IV; M/T ≥ 0.33 all stages	6× COPP/ABV+RT: - CR or PR ≥ 75%: 20 Gy IF - PR <75%: 36 Gy IF	4× COPP/ABV+: - CR: 2× COPP/ABV + RT 14.4 Gy IF - PR: 2× IEP + RT - CR: 14.4 Gy IF - PR: 25.2 Gy IF
If RP ≤ 50% after 2° cycle: GR1: IEP/OPPA/COPP/IEP + RT; GR2 and 3: IEP/OPPA/IEP/OPPA/IEP + RT		
ABVD Doxorubicin: 25 mg/m^2^ IV days 1 Bleomycin: 10 mg/m^2^ IV days 1 and 15 Vinblastine: 6 mg/m^2^ IV days 1 and 15 DTIC: 375 mg/m^2^ IV days 1 and 15	COPP/ABV Cyclophosphamide: 600 mg/m^2^ IV day 1 Vincristine: 1.4 mg/m^2^ IV day 1 Prednisone: 40 mg/m^2^ orally days 1–14 Procarbazine: 100 mg/m^2^ orally days 1–7 Doxorubicin: 35 mg/m^2^ IV day 8 Bleomycin: 10 mg/m^2^ IV day 8 Vinblastine: 6 mg/m^2^ IV day 8	**IEP** Ifosfamide: 2000 mg/m^2^ IV days 1–5 Etoposide: 120 mg/m^2^ IV days 1–5 Prednisone: 100 mg/m^2^ orally days 1–5
OPPA Vincristine: 1.5 mg/m^2^ IV days 1 and 8 and 15 Procarbazine: 100 mg/m^2^ orally days 1–14 Prednisone: 60 mg/m^2^ IV days 1–14 Doxorubicin: 40 mg/m^2^ IV days 1 and 15	COPP Cyclophosphamide: 500 mg/m^2^ IV days 1 and 8 Vincristine: 1.5 mg/m^2^ IV days 1 and 8 Procarbazine: 100 mg/m^2^ orally days 1–15 Prednisone: 40 mg/m^2^ orally days 1–15	

A: symptoms A (absent), B: symptoms B (present). CR: complete response. IF: involved fields. M/T: mediastinal/thoracic ratio. PR: partial response. RT: radiotherapy.

**Table 2 cancers-14-01471-t002:** Therapeutic regimens utilized as a second line treatment for cHL.

Acronyms	Drugs				
IEP	High-dose Ifosfamide	Etoposide	Metil-Prednisolone	-	-
DHAP	Dexamethasone	High dose Cytosine-Arabinoside	Cisplatin	-	-
DECAL	Dexamethasone	Etoposide	Cisplatin	Cytosine-Arabinoside	L-asparaginase
BEAM	BCNU	Etoposide	Cytosine-Arabinoside	Melphalan	-

**Table 3 cancers-14-01471-t003:** Characteristics at diagnosis of 272 recurring patients with classical Hodgkin’s lymphoma.

Patient Characteristics	*n*	%
Protocol		
MH 96	82	30.2
LH 2004	190	69.8
Gender		
Male	158	58.1
Female	114	41.9
Age (years)		
<5	3	1.1
5–14	180	66.2
≥15	89	32.7
Histology		
Lymphocyte depleted	5	1.8
Mixed cellularity	24	8.8
Nodular sclerosis	243	89.3
Stage		
1	7	2.6
2	125	46.0
3	62	22.8
4	78	28.7
Symptoms		
A	111	40.8
B	161	59.2
Bulky		
No	97	35.7
Yes	175	64.3
Number of involved sites		
1–3	64	23.5
4–7	116	42.7
≥8	92	33.8
Extra-nodal site involvement		
No	178	65.4
Yes	94	34.6
Treatment group		
1	12	4.4
2	35	12.9
3	225	82.7
Radiotherapy		
No according to protocol	6	2.2
No for disease progression	68	25.0
Yes	195	71.7
Missing	3	1.1

Extra-nodal sites: parenchymal sites, including liver, lung, bone, and bone marrow. %: percentages calculated on valid data only.

**Table 4 cancers-14-01471-t004:** Characteristics at disease recurrence of 272 patients with classical Hodgkin’s lymphoma.

Patient Characteristics	n	%
Type of recurrence		
Progression	117	43.0
Relapse	155	57.0
Early relapse (3–12 months from OT)	79	51.0
Late relapse (≥12 months from OT)	76	49.0
Age (years)		
<5	3	1.1
5–14	121	44.5
≥5	148	54.4
Stage		
1	30	12.6
2	91	38.2
3	36	15.1
4	81	34.0
Missing	34	12.5
Number of involved sites *		
1	72	29.8
2–4	102	42.5
≥5	68	28.1
Missing	30	11.0
Extra-nodal site involvement **		
No	156	64.7
Yes	85	35.3
Missing	31	11.4
Recurrence at the same site		
No	41	16.9
Yes	201	83.1
Missing	30	11.0
Recurrence after Radiotherapy		
Recurrence in non-irradiated sites	30	16.7
Recurrence in the irradiated site	150	83.3
Missing	15	7.7

* The number of involved sites was grouped according to the tertiles of its distribution. ** Extra-nodal sites: parenchymal sites including liver, lung, bone, and bone marrow. % percentages calculated on valid data only.

**Table 5 cancers-14-01471-t005:** Ten-year overall survival of 272 patients with classical Hodgkin’s lymphoma after recurrence by characteristics at diagnosis.

Patient Characteristics	N/D	OS	95% CI	*p*
Whole cohort	272/89	65.3	59.0–70.9	
Protocol				**0.042**
MH96	82/36	56.5	45.0–66.5	
LH2004	190/53	69.5	61.7–76.0	
Gender				0.687
Male	158/53	62.3	53.2–70.2	
Female	114/36	66.9	56.7–75.3	
Age (years)				0.729
0–14	183/62	64.5	56.8–71.2	
≥15	89/27	66.6	53.9–76.5	
Histology				0.356
Lymphocyte depleted	5/2	50.0	5.8–84.5	
Mixed cellularity	24/10	57.4	35.2–74.4	
Nodular sclerosis	243/77	66.4	59.7–72.3	
Stage				0.885
1–2	132/44	65.5	56.2–73.2	
3–4	140/45	65.3	56.2–73.0	
Symptoms				0.052
A	111/29	72.0	61.8–80.0	
B	161/60	60.8	52.5–68.2	
Bulky disease				0.141
No	97/26	71.2	60.0–79.8	
Yes	175/63	62.3	54.4–69.2	
Number of involved sites *				0.667 *
1–3	64/22	65.6	52.2–76.1	
4–7	116/39	64.1	54.1–72.5	
≥8	92/28	66.4	54.8–75.7	
Extra-nodal site involvement **				0.220
No	178/62	63.7	55.8–70.5	
Yes	94/27	68.6	57.4–77.5	
Treatment Group				0.220
1	12/3	83.3	48.2–95.6	
2	35/7	76.9	56.9–88.5	
3	225/79	62.6	55.5–68.8	
Radiotherapy (post-chemotherapy)				0.197
No according to protocol	6/0	100		
No for disease progression	68/25	62.3	49.5–72.8	
Yes	195/63	65.2	57.5–71.9	

* The number of involved sites was grouped according to the tertiles of its distribution. ** Extra-nodal sites: parenchymal sites including liver, lung, bone, and bone marrow. % percentages calculated on valid data only. Bold: significant *p* value.

**Table 6 cancers-14-01471-t006:** Ten-year overall survival of 272 patients with classical Hodgkin’s lymphoma after recurrence by characteristics at relapse/progression.

Patient Characteristics	N/D	OS	95% CI	*p*
Type of recurrence				**<0.001**
Progression	117/54	52.1	42.5–61.0	
Relapse	155/35	75.3	66.9–81.8	
Early relapse	79/18	73.7	60.9–82.9	0.991
Late relapse	76/17	76.5	64.3–85.0	
Age (years)				0.252
0–14	124/46	61.5	52.0–69.6	
≥15	148/43	68.7	59.7–76.1	
Stage				**0.029**
1–2	121/36	69.0	59.6–76.7	
3–4	117/48	57.2	47.1–66.1	
Number of involved sites *				**0.023** *
1	72/22	68.4	56.0–78.0	
2–4	102/32	65.8	55.0–74.7	
≥5	68/32	52.5	39.3–64.2	
Extra-nodal site involvement **				0.087
No	156/51	66.9	58.7–73.8	
Yes	85/35	54.9	42.7–65.5	
Recurrence at the same site				0.412
No	41/13	66.4	49.0–79.0	
Yes	201/73	62.3	54.9–68.9	
Recurrence after Radiotherapy				0.110
Recurrence in a non-irradiated site	30/7	75.7	55.5–87.6	
Recurrence in the same irradiated site	150/54	61.8	52.8–69.5	

* The number of involved sites was grouped according to the tertiles of its distribution. ** Extra-nodal sites: parenchymal sites including liver, lung, bone, and bone marrow. % percentages calculated on valid data only. Bold: significant *p* value.

**Table 7 cancers-14-01471-t007:** Ten-year event-free survival of 272 patients with classical Hodgkin’s lymphoma after recurrence by characteristics at relapse/progression.

Patient Characteristics	N/E	EFS	95% CI	*p*
Type of recurrence				**<0.001**
Progression	117/69	38.0	28.6–47.4	
Relapse	155/52	64.6	56.0–71.9	
Early relapse	79/30	59.7	47.3–70.0	0.244
Late relapse	76/22	69.7	57.2–79.3	
Age (years)				0.571
0–14	124/59	51.6	42.2–60.3	
≥15	148/62	54.3	45.1–62.7	
Stage				**0.011**
1–2	121/48	59.3	49.8–67.6	
3–4	117/64	42.3	32.4–51.8	
Number of involved sites *				0.079 *
1	72/31	56.4	44.0–67.0	
2–4	102/45	52.4	41.2–62.4	
≥5	68/38	43.6	31.6–55.0	
Extra-nodal site involvement **				**0.019**
No	156/67	56.5	48.1–64.0	
Yes	85/47	40.0	28.4–51.3	
Recurrence at the same site				0.331
No	41/18	56.8	39.8–70.7	
Yes	201/96	49.8	42.3–56.9	
Recurrence after radiotherapy				0.067
Recurrence in a non-irradiated site	30/10	69.4	49.4–82.8	
Recurrence in the same irradiated site	150/70	50.0	41.0–58.3	

* The number of involved sites was grouped according to the tertiles of its distribution. ** Extra-nodal sites: parenchymal sites including liver, lung, bone, and bone marrow. % percentages calculated on valid data only. Bold: significant *p* value.

**Table 8 cancers-14-01471-t008:** Univariable and multivariable Cox regression model to assess the overall survival after recurrence by characteristics at diagnosis and at relapse/progression of 272 patients with classical Hodgkin’s lymphoma.

		Univariable Analysis			Multivariable Analysis		
Patient Characteristics	N/D	HR	95% CI	*p*	HR	95% CI	*p*
Characteristics at diagnosis							
Protocol				**0.042**			**<0.001**
MH96	82/36	1 (ref)	-		1 (ref)	-	
LH2004	190/53	0.65	0.42–0.99		0.43	0.27–0.70	
Symptoms at diagnosis				**0.054**			0.184
A	111/29	1 (ref)	-		1 (ref)	-	
B	161/60	1.5	0.99–2.4		1.4	0.86–2.2	
Characteristics at recurrence							
Type of recurrence				**<0.001**			**<0.001**
Progression	117/54	1 (ref)	-		1 (ref)	-	
Relapse	155/35	0.38	0.25–0.58		0.33	0.21–0.52	
Stage				**0.029**			0.147
1–2	121/36	1 (ref)	-		1 (ref)	-	
3–4	117/48	1.6	1.0–2.5		1.4	0.88–2.3	
Extra-nodal site involvement *				**0.093**			0.409
No	156/51	1 (ref)	-		1 (ref)	-	
Yes	85/35	1.5	0.95–2.2		1.2	0.76–2.0	
Number of involved sites **				**0.054**			**0.009**
1	72/22	1 (ref)	-		1 (ref)	-	
2–4	102/32	1.1	0.66–1.9		1.5	0.85–2.5	
≥5	68/32	1.9	1.1–3.2		2.5	1.4–4.4	
Continuous variable	242/86	1.1	1.04–1.2	**0.003**	1.1	1.1–1.2	**0.001**

N/D = Number of patients/deaths. HR = hazard ratio. * Extra-nodal sites: parenchymal sites including liver, lung, bone, and bone marrow. % percentages calculated on valid data only. ** The number of involved sites was grouped according to the tertiles of its distribution. Bold: significant *p* value.

**Table 9 cancers-14-01471-t009:** Univariable and multivariable Cox regression model to assess the event-free survival after recurrence by characteristics at diagnosis and at relapse/progression of 272 patients with classical Hodgkin’s lymphoma.

		Univariable Analysis			Multivariable Analysis		
Patient Characteristics	N/E	HR	95% CI	*p*	HR	95% CI	*p*
Characteristics at diagnosis							
Protocol				**0.099**			**0.006**
MH96	82/45	1 (ref)	-		1 (ref)	-	
LH2004	190/76	0.73	0.50–1.1		0.56	0.38–0.84	
Symptoms				**0.067**			0.200
A	111/42	1 (ref)	-		1 (ref)	-	
B	161/79	1.4	0.98–2.1		1.3	0.87–1.9	
Characteristics at recurrence							
Type				**<0.001**			**<0.001**
Progression	117/69	1 (ref)	-		1 (ref)	**-**	
Relapse	155/52	0.43	0.30–0.61		0.34	0.23–0.51	
Stage				**0.012**			**0.005**
1–2	121/48	1 (ref)	-		1 (ref)	**-**	
3–4	117/64	1.6	1.1–2.4		1.7	1.2–2.5	
Number of involved sites *				**0.072**			0.492
Continuous variable	242/114	1.06	1.0–1.12		1.02	0.96–1.10	
Extra-nodal site involvement **				**0.022**			0.783
No	156/67	1 (ref)	-		1 (ref)	-	
Yes	85/47	1.6	1.1–2.3		1.1	0.62–1.9	
Recurrence after radiotherapy ***				**0.052**			**0.031**
Non-irradiated site	30/10	1 (ref.)	-		1 (ref.)	-	
Same irradiated site	150/70	1.8	0.95–3.6		2.0	1.0–4.1	

N/E: Number of patients/events; HR = hazard ratio. Extra-nodal sites: parenchymal sites, including liver, lung, bone, and bone marrow. * The number of involved sites was grouped according to the tertiles of its distribution ** Extra-nodal sites: parenchymal sites including liver, lung, bone, and bone marrow. % percentages calculated on valid data only. *** Analysis performed in the subcohort of patients receiving radiotherapy after first-line chemotherapy (*n* = 174). Bold: significant *p* value.

## Data Availability

The Database is in CINECA, which is a Consortium composed by 102 Italian universities and public institutions. Since its origins in 1969, Cineca offers support to scientific research, public and industrial, through supercomputing and the use of the most innovative computing systems based on state-of-the-art architectures and technologies. The database is not public, as it is affirmed in the informed consent signed by patients or parents.

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
