# Peer review of "Outcome of Children and Adolescents with Recurrent Classical Hodgkin Lymphoma: The Italian Experience"

_cancers, 2022, doi:10.3390/cancers14061471_

Round 1
Reviewer 1 Report
In this paper, the authors studied a wide population of children and adolescents with relapsed/refractory Hodgkin lymphoma previously enrolled in two subsequent Italian protocols, and identified risk factors associated to worse OS and EFS.
The paper is well written and methodically correct and the data are interesting and convincing.
The heterogeneous type of utilized treatments at relapse, the retrospective nature of the analysis and the wide time interval of evaluation with an inconstant use of FDG-PET for response assessment may represent a limit of this study. However, results remain statistically significant and scientifically valid.
Discussion is appropriate and final clinical recommendation of a tailored approach to treatment of refractory/relapsed Hodgkin lymphoma in children and adolescents are strong and in agreement with previous studies.
Minor comments: table 5 may be omitted (or included among supplement material) because it reports no significant data.
Author Response
In this paper, the authors studied a wide population of children and adolescents with relapsed/refractory Hodgkin lymphoma previously enrolled in two subsequent Italian protocols, and identified risk factors associated to worse OS and EFS.
The paper is well written and methodically correct and the data are interesting and convincing.
The heterogeneous type of utilized treatments at relapse, the retrospective nature of the analysis and the wide time interval of evaluation with an inconstant use of FDG-PET for response assessment may represent a limit of this study. However, results remain statistically significant and scientifically valid.
Discussion is appropriate and final clinical recommendation of a tailored approach to treatment of refractory/relapsed Hodgkin lymphoma in children and adolescents are strong and in agreement with previous studies.
We thank the Reviewer for his/her appreciation of our study.
Minor comments: table 5 may be omitted (or included among supplement material) because it reports no significant data.
Following the Reviewer’s suggestion the table have been moved to the Supplemental Material section (Supplemental Table 3)

Reviewer 2 Report
The authors have made a major effort to integrate retrospective clinical data from children and adolescent patients with classical Hodgkin's lymphoma, aimed at identifying prognostic factors for relapsed or progressive disease and improving patient stratification strategies. Congratulations to the authors.
The following are some suggestions that may help improve the readability and presentation of the results, and should be addressed before acceptance of the manuscript.
Structure:
- Figures 1 and 2 should be treated as tables and kept in the main manuscript.
- At the editor's discretion. I believe that tables 1-8 are necessary, but should be presented as supplementary material.
- Schemes 1 and 2 (in the supplementary material) should be renamed as supplementary tables X and Y and should be cited in the text.
- The captions for each figure/table should cover all the abbreviations included. Please, add or revise all captions.
Minor typos:
- Line 178: LH, HL or cHL?
- Lines 175-186 present data that are difficult to read. Include them in a supplementary table or so, please.
- Line 205: Add the word “significantly”.
- The captions for tables 7 and 8 should be improved because it is not clear until the end of the legend what type of analysis has been performed.
- Funding statement. At the editor’s discretion. Reference numbers for each funding should be indicated.
Other comments and opinions:
- The manuscript could be more educational by describing some clinical terms, e.g. Symptoms A and B, terms in Figure 2, etc.
- If tables are presented in the supplementary material, specify the most important results in the text, including significant p-values or relevant data.
- Please, revise the values included in the text because there are some mistakes (line 151, 69.8% instead of 70%).
- These clinical data are relevant and very valuable, especially in paediatric and young populations. Currently, liquid biopsy-based biomarkers, such as cell-free DNA (PMID: 31112884), have proven useful in predicting relapse or treatment failure in lymphomas and are a good complement to clinical data. This should be highlighted in the discussion. Furthermore, in infants, who experience several developmental events over years (PMID: 31112884), the utility of biomarkers should be evaluated before their clinical implementation.
- A good final message might be to recommend accumulation of biomarker data in the next prospective cohort.
Author Response
The authors have made a major effort to integrate retrospective clinical data from children and adolescent patients with classical Hodgkin's lymphoma, aimed at identifying prognostic factors for relapsed or progressive disease and improving patient stratification strategies. Congratulations to the authors.
We thank the Referee for her/his appreciation of our research.
The following are some suggestions that may help improve the readability and presentation of the results, and should be addressed before acceptance of the manuscript.
Structure:
- Figures 1 and 2 should be treated as tables and kept in the main manuscript.
Following the Reviewer’s advice Figure 1 and 2 have been transformed into tables and numbered accordingly.
- At the editor's discretion. I believe that tables 1-8 are necessary, but should be presented as supplementary material.
Table 5 have been moved to Supplemental Material also following the suggestions of the other Referee. With regards to the other tables we will conform to the Editor’s suggestions.
- Schemes 1 and 2 (in the supplementary material) should be renamed as supplementary tables X and Y and should be cited in the text.
The two tables in Supplemental Material have been renamed and they are cited in the results section.
- The captions for each figure/table should cover all the abbreviations included. Please, add or revise all captions.
Table and figure legends have been integrated and modified to include all the abbreviations used.
Minor typos:
- Line 178: LH, HL or cHL?
We thank the Referee for his/her comment. The mistakes have been corrected.
- Lines 175-186 present data that are difficult to read. Include them in a supplementary table or so, please.
A flow chart has been added as Supplemental Figure 1 to better illustrate the different outcomes of the 272 studied patients and the related explanation in the Results section rephrased accordingly.
- Line 205: Add the word “significantly”.
The word “significantly” has been inserted before the word “poorer” at the line 205.
- The captions for tables 7 and 8 should be improved because it is not clear until the end of the legend what type of analysis has been performed.
The captions of Figure 7 and 8 have been rephrased following the Reviewer’s advice.
- Funding statement. At the editor’s discretion. Reference numbers for each funding should be indicated.
No reference code is associated to the "Cinque per mille" and "Ricerca corrente" grant, since this funding is given by the Italian Health Ministry to all research Institutes.
Other comments and opinions:
- The manuscript could be more educational by describing some clinical terms, e.g. Symptoms A and B, terms in Figure 2, etc.
It was done
- If tables are presented in the supplementary material, specify the most important results in the text, including significant p-values or relevant data.
The most relevant results are now reported in the main text of the manuscript.
- Please, revise the values included in the text because there are some mistakes (line 151, 69.8% instead of 70%).
The manuscript has been thoroughly revised and the detected mistakes have been corrected.
- These clinical data are relevant and very valuable, especially in paediatric and young populations. Currently, liquid biopsy-based biomarkers, such as cell-free DNA (PMID: 31112884), have proven useful in predicting relapse or treatment failure in lymphomas and are a good complement to clinical data. This should be highlighted in the discussion. Furthermore, in infants, who experience several developmental events over years (PMID: 31112884), the utility of biomarkers should be evaluated before their clinical implementation.
- A good final message might be to recommend accumulation of biomarker data in the next prospective cohort.
A paragraph regarding biomarkers was written as well as final message.
